# Biomimetic non-classical crystallization drives hierarchical structuring of efficient circularly polarized phosphors

Li-Zhe Feng[1,2], Jing-Jing Wang[1,2], Tao Ma[3], Yi-Chen Yin[1,2], Kuang-Hui Song[1,2], Zi-Du Li[4,5,6], Man-Man Zhou[7], Shan Jin[7], Taotao Zhuang[1,3], Feng-Jia Fan [4,5,6], Man-Zhou Zhu [7] & Hong-Bin Yao [1,2✉]

Hierarchically structured chiral luminescent materials hold promise for achieving efficient circularly polarized luminescence. However, a feasible chemical route to fabricate hierarchically structured chiral luminescent polycrystals is still elusive because of their complex structures and complicated formation process. We here report a biomimetic non-classical crystallization (BNCC) strategy for preparing efficient hierarchically structured chiral luminescent polycrystals using well-designed highly luminescent homochiral copper(I)-iodide hybrid clusters as basic units for non-classical crystallization. By monitoring the crystallization process, we unravel the BNCC mechanism, which involves crystal nucleation, nanoparticles aggregation, oriented attachment, and mesoscopic transformation processes. We finally obtain the circularly polarized phosphors with both high luminescent efficiency of 32% and high luminescent dissymmetry factor of $1.5 \times 10^{-2}$, achieving the demonstration of a circularly polarized phosphor converted light emitting diode with a polarization degree of 1.84% at room temperature. Our designed BNCC strategy provides a simple, reliable, and large-scale synthetic route for preparing bright circularly polarized phosphors.

[1] Hefei National Research Center for Physical Sciences at the Microscale, University of Science and Technology of China, Hefei, Anhui 230026, China. [2] Department of Applied Chemistry, Hefei Science Center of Chinese Academy of Sciences, University of Science and Technology of China, Hefei, Anhui 230026, China. [3] Department of Chemistry, Hefei Science Center of Chinese Academy of Sciences, University of Science and Technology of China, Hefei, Anhui 230026, China. [4] Department of Physics, Hefei Science Center of Chinese Academy of Sciences, University of Science and Technology of China, Hefei, Anhui 230026, China. [5] CAS Key Laboratory of Microscale Magnetic Resonance, University of Science and Technology of China, Hefei, Anhui 230026, China. [6] Synergetic Innovation Center of Quantum Information and Quantum Physics, University of Science and Technology of China, Hefei, Anhui 230026, China. [7] Key Laboratory of Structure and Functional Regulation of Hybrid Materials, Anhui University, Ministry of Education, Hefei, Anhui 230601, China. ✉email: yhb@ustc.edu.cn

In recent years, chiroptical materials with circularly polarized luminescence (CPL) characteristics have received considerable attention in the fields of photoelectric devices[1], 3D optical displays[2,3], optical information encryptions[4,5], and optical sensors[6]. The performance of CPL-active materials can be evaluated by the luminescent efficiency and luminescence dissymmetry factor (i.e. $g_{lum}$), which is defined as

$$g_{lum} = 2\frac{I_L - I_R}{I_L + I_R} \quad (1)$$

where $I_L$ and $I_R$ are intensities of left- and right-handed circularly polarized emissions. The maximum value of $g_{lum}$ is $\pm 2$, which implies the ideal left- or right-handed circularly polarized luminescence. However, because of the inter-restriction of magnetic dipole transition and electric dipole transition, traditional CPL-active materials, such as chiral luminescent organic molecules and coordination compounds, are difficult to obtain both high photoluminescence efficiency and large $g_{lum}$ values[6,7].

The hierarchical configurations of chiroptical units are promising to enhance CPL performance via structural chirality amplification through multiscale levels[8,9]. Some significant progresses have been made in hierarchical construction of chiral organic assemblies for efficient CPL performance with the $g_{lum}$ of $10^{-2}$–$10^{-1}$ magnitude[10,11]. Extending the organic assemblies to organic-inorganic hybrid chiral system will broaden the scope to explore the CPL-active materials based on the chiral aggregation-induced emission units and macrostructures. Additionally, it will enrich the CPL-active material sources to understand the mechanisms of structure-function relationships between the assembled hierarchical structures and chiral units. Therefore, an advanced strategy to conduct hierarchically structured assemblies of organic-inorganic hybrid chiral luminescent units across multiscale levels towards efficient CPL is desirable[8,12,13].

In natural biomaterials, hierarchical structures regulated by chiral molecules across multiple scales for chirality amplification are considerably ubiquitous. As the building blocks of lives on the earth, chiral sugars and amino acids endow the biomacromolecules such as nucleic acids and proteins with stable helical structures[14]. These macromolecules like proteins serve as an organic matrix, which induces and regulates the stereoselective non-classical crystallization of calcium carbonate at room temperature to form the macroscale chiral-shaped biominerals[15,16], e.g., helical textures on seashells[17]. This implies that the hierarchically structured chiral polycrystals can be synthesized for chirality amplification through a feasible biomimetic non-classical crystallization (BNCC) process at room temperature based on the well-designed homochiral units[18]. Nevertheless, using the mild BNCC strategy to construct artificial hierarchically structured polycrystals with high-performance CPL has never been achieved due to the absence of homochiral luminescent units.

An appropriate homochiral unit to construct hierarchical CPL-active assemblies via BNCC should be structurally robust and highly luminescent. Here, we turn our attention to copper(I)-iodide (Cu-I) hybrid clusters owing to their highly luminescent efficiency characteristics, as well as structural identity and robustness for the rational design of homochiral units[19–26]. We consider that the chiral Cu-I hybrid clusters are potential homochiral units to fabricate hierarchically structured CPL-active materials through the feasible and mild BNCC route.

Herein, we report a BNCC route to synthesize hierarchical structures induced chirality amplified CPL-active materials as efficient circularly polarized phosphors endowing light emitting diodes (LEDs) bright circularly polarized emissions. Our strategy is featured by the multiscale design and assembly process mimicking the formation of chiral biominerals. We design and synthesize a series of chiral Cu-I hybrid clusters as highly luminescent homochiral units. Thereafter, to amplify the chirality for efficient CPL, we conduct the assemblies of Cu-I hybrid clusters at nano- and micro-scales to yield hierarchically structured CPL polycrystals via the BNCC. Compared to directly precipitated powders without hierarchical chiral structures, our hierarchical assemblies of chiral Cu-I hybrid clusters exhibit the much higher $g_{lum}$ value of $1.5 \times 10^{-2}$, as well as maintaining high photoluminescence quantum yield (32%). Using obtained polycrystals as phosphors, we present a CPL-active light emitting diode device with a polarization degree of 1.84% at room temperature.

## Results and discussion

**Homochiral luminescent cluster design and synthesis.** Aiming to the rational design of reliable basic units, we select chirally modified triethylenediamine (ted) as chiral ligands to construct robust and luminescent homochiral Cu-I hybrid clusters. Synthetic procedures of these four chiral ligands are described in Fig. 1a. To study the effects of chiral center position and chain length of ligands on the BNCC assembly, four chiral ligands with different alkyl chain lengths and chiral center positions are designed (Fig. 1b). Four different chiral iodohydrocarbons are synthesized from corresponding chiral alcohol molecules by nucleophilic substitution (detail phase see Methods)[27]. Then, such chiral alkyl chains are introduced into ted to form chiral quaternary ammonium salts, which serve as ligands combined with Cu-I inorganic modules using all-in-one typed coordinate bonds and ionic bonds together to construct chiral hybrid clusters[23]. All the synthetic organic compounds can be confirmed by $^1H$ NMR and $^{13}C$ NMR analysis (Supplementary Figs. 1–8).

As shown by fluorescence microscopy images (Fig. 2a–d), high-quality single crystals of chiral clusters were obtained by the slow diffusion method[23,24,28]. All chiral cluster structures in the obtained single crystals can be determined by single-crystal X-ray diffraction (SCXRD). As shown in Fig. 1c, the obtained chiral clusters from $L_2$ to $L_4$ have the same Cu-I nucleus, and the corresponding formulas can be illustrated as $Cu_5I_7(L_2)_2$, $Cu_5I_7(L_3)_2$, and $Cu_5I_7(L_4)_2$, respectively. Differently, the crystal obtained by $L_1$ is constructed by $Cu_4I_4(L_1)_4^{4+}$ as cationic clusters and one-dimensional $Cu_8I_{12}^{4-}$ chains as counterparts to balance the charges. Four kinds of hybrid clusters show the specific spatial arrangements in the crystal lattices (Fig. 1d and Supplementary Figs. 10–13). Detailed crystallography parameters of the obtained crystals are shown in Supplementary cif. files and summarized in Supplementary Table 1. Besides, powder X-ray diffraction (PXRD) patterns of four directly solution-precipitated powders are highly consistent with the simulated ones, which can verify their high crystallinity and phase purity (Supplementary Fig. 9). In addition, all these clusters crystallize into chiral space groups, namely, $P4_2$, $P1$, $P3_221$, and $P1$, corresponding to $Cu_4I_4(L_1)_4^{4+}$ $Cu_8I_{12}^{4-}$, $Cu_5I_7(L_2)_2$, $Cu_5I_7(L_3)_2$, and $Cu_5I_7(L_4)_2$, respectively. Meanwhile, there are 4-fold and 3-fold screw axes, which along the c axes in $Cu_4I_4(L_1)_4^{4+}$ $Cu_8I_{12}^{4-}$ and $Cu_5I_7(L_3)_2$ crystal lattices, respectively (Supplementary Figs. 14 and 15). However, no symmetry elements can be found in $Cu_5I_7(L_2)_2$ and $Cu_5I_7(L_4)_2$ crystal lattices because of the $P1$ space group. Additionally, angle-dependent circularly dichroism (CD) signals of these fully ground isotropic powders measured by diffuse reflectance are clearly distinct (Supplementary Fig. 16), implying that the chirality was successfully introduced into these Cu-I hybrid clusters on the molecular scale.

**Optical properties of chiral cluster based single-crystal powders.** To explore the optical properties of the synthesized Cu-I hybrid chiral clusters, a series of characterizations on these

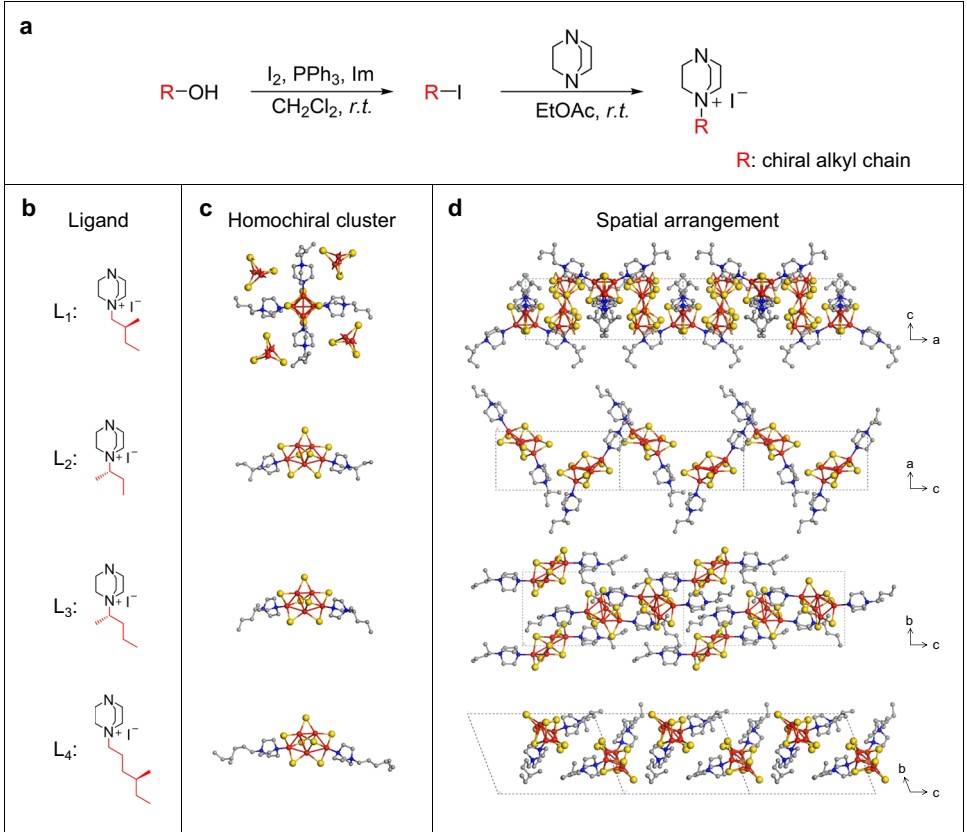

**Fig. 1 Chiral ligand design and structure characteristics of the synthesized chiral clusters. a** Reaction procedure scheme for the synthesis of four chiral ligands ($L_1$ to $L_4$). Im, Imidazole; *r.t.*, room temperature. **b** structures of four chiral ligands $L_1$-$L_4$. The highlighted structure segments in red color are refer to the chiral alkyl chain R in **a**. **c** Four chiral Cu-I hybrid clusters $Cu_4I_4(L_1)_4^{4+}$ $Cu_8I_{12}^{4-}$, $Cu_5I_7(L_2)_2$, $Cu_5I_7(L_3)_2$ and $Cu_5I_7(L_4)_2$. Green shadings, coordination site; blue shadings, chiral center. **d** The corresponding schematics of spatial arrangements along different axes in crystal lattices for hybrid clusters $Cu_4I_4(L_1)_4^{4+}$ $Cu_8I_{12}^{4-}$, $Cu_5I_7(L_2)_2$, $Cu_5I_7(L_3)_2$ and $Cu_5I_7(L_4)_2$. The crystal structures are obtained by crystallography. Color scheme: Cu, red; I, yellow; N, blue; C, Gray.

synthesized single-crystal powders were carried out. Figure 2e shows steady photoluminescent (PL) spectra and UV–vis absorption spectra of these crystal powders. The PL peaks of four crystal powders shift from green (540 nm) to orange-red (650 nm) with the chiral ligand employed from $L_1$ to $L_4$. Alkyl chain lengths show an obvious influence on the luminescent efficiency as that the photoluminescence quantum yields (PLQY) increase with shortening the chain length (50%, 65%, 10%, 7% for $Cu_4I_4(L_1)_4^{4+}$ $Cu_8I_{12}^{4-}$, $Cu_5I_7(L_2)_2$, $Cu_5I_7(L_3)_2$, $Cu_5I_7(L_4)_2$, respectively). It is probably because the shorter alkyl chains have fewer rotations and wiggles in the crystal lattice to enhance the aggregation-induced light emission[29,30]. UV–vis absorption spectra show that all these clusters have strong absorptions of light at the wavelength <400 nm, implying their potentials as efficient phosphors due to large Stokes shifts induced weak self-absorption. Thermogravimetric analyses (TGA) further indicate that the obtained hybrid clusters have excellent thermal stabilities with average thermal decomposition temperatures >200 °C (Fig. 2f). In addition, PL lifetimes decay curves of $Cu_4I_4(L_1)_4^{4+}$ $Cu_8I_{12}^{4-}$ and $Cu_5I_7(L_2)_2$ show the best fit with single (first-order) exponential decay functions (Fig. 2g), and the PL lifetime decay constant ($\tau$) is 7.2 and 3.4 μs respectively, identifying that their emission behaviors belong to phosphorescence, which can be described as that exited electrons in the singlet-exited state ($S_{>1}$) pass through the internal conversion (IC) and intersystem crossing (ISC) to the lowest triplet-exited state ($T_1$), and then through radiative recombination with the holes in the ground state ($S_0$) (Fig. 2h). In addition, these isotropic crystal powders

behave CPL, but their $g_{lum}$ values were still low on the order of $10^{-4}$ magnitude (Supplementary Fig. 17).

**BNCC of hierarchically organized chiral polycrystals**. As shown in the above results, CD and CPL signals of single-crystal powders are relatively weak. Aiming to largely improve chiroptical properties, we attempt to magnify the chirality of the well-designed chiral clusters via the BNCC induced hierarchically structural assemblies (Fig. 3a). We used polyvinylpyrrolidone (PVP, K88-96), a kind of amphiphilic polymer, to confine Cu-I ionic chains in the micelle, which formed a homogeneous colloidal solution according to our previous work[31]. Then, the synthesized chiral ligands were added into the colloidal solution system, in which the coordinative and electrostatic interactions make these primary liquid precursors combine together to form the designed chiral clusters. Under the regulation of PVP and excess chiral ligands, the aggregation of chiral clusters gradually transformed into hierarchically structured polycrystals through the BNCC process.

We monitored the crystallization process of $Cu_4I_4(L_1)_4^{4+}$ $Cu_8I_{12}^{4-}$ and $Cu_5I_7(L_3)_2$ hierarchical polycrystals at different crystal formation states by scanning electron microscopy (SEM) and transmission electron microscopy (TEM). Impressively, the crystallization process can be well described by BNCC, which is classified in the following four stages. (1) Crystal nucleation: intermediary clusters composed of inorganic and organic units act as building blocks to form crystallized nanoparticles through

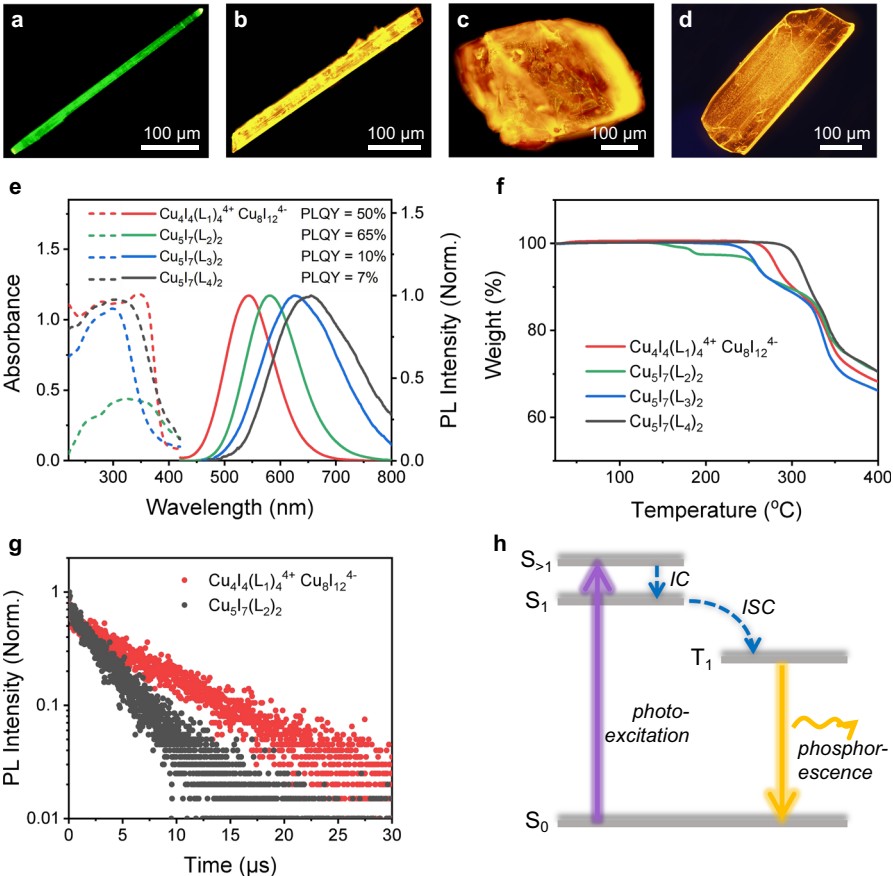

**Fig. 2 Optical properties, thermal stabilities, and photoluminescence mechanism of the synthesized hybrid crystal powders. a–d** Fluorescence microscopy images of $Cu_4I_4(L_1)_4^{4+}$ $Cu_8I_{12}^{4-}$ (**a**), $Cu_5I_7(L_2)_2$ (**b**), $Cu_5I_7(L_3)_2$ (**c**) and $Cu_5I_7(L_4)_2$ (**d**) single crystals. **e** Steady PL (solid) spectra and UV–vis absorption (dotted) spectra and of $Cu_4I_4(L_1)_4^{4+}$ $Cu_8I_{12}^{4-}$, $Cu_5I_7(L_2)_2$, $Cu_5I_7(L_3)_2$, and $Cu_5I_7(L_4)_2$ single-crystal powders at room temperature. PLQY, the photoluminescence quantum yields. The PLQY of single-crystal powders is 50%, 65%, 10%, 7% for $Cu_4I_4(L_1)_4^{4+}$ $Cu_8I_{12}^{4-}$, $Cu_5I_7(L_2)_2$, $Cu_5I_7(L_3)_2$ and $Cu_5I_7(L_4)_2$, respectively. **f** Corresponding TGA curves of four kinds of hybrid crystals. **g** PL lifetime decay curves of $Cu_4I_4(L_1)_4^{4+}$ $Cu_8I_{12}^{4-}$ and $Cu_5I_7(L_2)_2$ hybrid crystals. **h** Photoluminescence mechanism of corresponding Cu-I hybrid crystals indicating the phosphorescence emission process. S singlet state, T triplet state, IC internal conversion, ISC intersystem crossing. Source data are provided as a Source Data file.

crystal nucleation in micelles, which can be revealed by SEM images (Fig. 3b, f) and powder X-ray diffraction (PXRD) patterns (Supplementary Fig. 18a, c) show the consistent diffraction peaks with that of single-crystal powders of $Cu_4I_4(L_1)_4^{4+}$ $Cu_8I_{12}^{4-}$ and $Cu_5I_7(L_3)_2$. (2) Nanoparticles aggregation: as reactions progress, nanoparticles aggregate together through collision among nanoparticles and have a tendency to form specific shapes driven by the lattice formation energy of crystallization (Fig. 3c, g). PXRD patterns show narrower widths of peaks than that of nanoparticles at the first stage, indicating that the hybrid crystal grain size became larger, confirming the aggregation of nanoparticles (Supplementary Fig. 18a, c). (3) Oriented attachment: the lattice formation energy, dipolar-dipolar interactions, and Van der Waals forces can further induce these aggregated nanoparticles to arrange with a specific orientation. At this stage, excess chiral ligands, regarded as a kind of amphipathic surfactants, were partially and dynamically absorbed on the nanoparticle surfaces to "code" their alignment through the Van der Waals forces[32]. More importantly, excess chiral ligands in the colloidal solution created a chiral microenvironment, inducing the helically oriented attachment of nanoparticles and spontaneously forming highly ordered attached-nanocrystals with primary helical morphology (Fig. 3d, h). (4) Mesoscopic transformation into polycrystals: at this stage, attached-nanocrystals have already exhibited crystallographic register because of the highly ordered

arrangement of nanoparticles. However, they behave transient features due to the high driving force of the fusion of crystal boundaries[32]. To be specific, although nanoparticles align with each other with high orientation, some gaps are still preserved among them. Enough rotational freedom still exists so that particles can rotate and wiggle to correct positions[33]. At one certain point, crystal faces are driven to mutually align and fuse by a large thermodynamic driving force[34,35]. Eventually, attached-nanocrystals evolved to the final hierarchically organized chiral polycrystals with high thermodynamic stability (Fig. 3e, i). Meanwhile, SAEDs patterns for different crystallization stages of $Cu_4I_4(L_1)_4^{4+}$ $Cu_8I_{12}^{4-}$ polycrystals show the transformation from diffraction rings to clear diffraction spots, corresponding to the transformation from amorphous nanoparticles to crystalline polycrystals, which can further prove the four different stages of the BNCC process (Supplementary Fig. 19).

We further confirm the driving forces of assembly and chiral regulation through the high-resolution transmission electron microscopy (HRTEM). As shown in Fig. 3j, the lattice fringe spacing is 3.7 Å and 7.8 Å, corresponding to the interplanar spacing (600) and (220) planes of the assembled $Cu_4I_4(L_1)_4^{4+}$ $Cu_8I_{12}^{4-}$ polycrystals, respectively. This reveals that the exposed crystal planes in the obtained polycrystals is the (002) plane dominantly, which further indicates that the helical arrangements of assembly units depend on a specific steric configuration caused

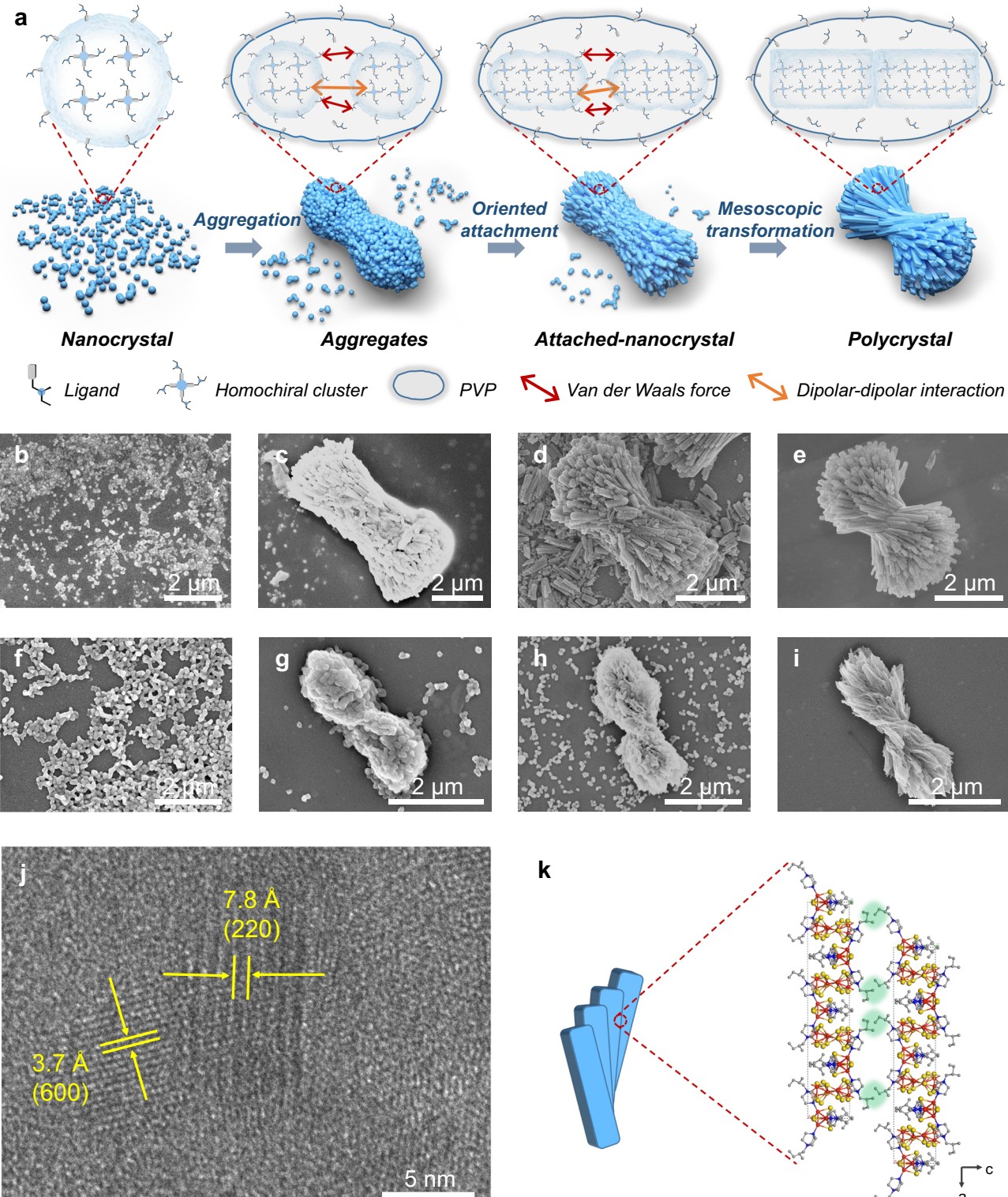

**Fig. 3 BNCC of hierarchically organized chiral polycrystals. a** Schematic diagram of biomimetic non-classical crystallization process involving nanocrystal nucleation, aggregation, oriented attachment and mesoscopic transformation. PVP is the amphiphilic polymer to confine Cu-I ionic chains and Cu-I hybrid clusters in the micelle. The $Cu_8I_{12}^{4-}$ chains were omitted for clarity. **b**–**e** SEM images of different crystallization stages of $Cu_4I_4(L_1)_4^{4+}$ $Cu_8I_{12}^{4-}$ polycrystals with reaction time of 3 min (**b**), 20 min (**c**), 1 h (**d**), 6 h (**e**), respectively ([$L_1$] = 6 mM). **f**–**i** SEM images of different crystallization stages of $Cu_5I_7(L_3)_2$ polycrystals with reaction time of 2 h (**f**), 4 h (**g**), 5 h (**h**), 6 h (**i**), respectively ([$L_3$] = 25 mM). **j** HRTEM image of $Cu_4I_4(L_1)_4^{4+}$ $Cu_8I_{12}^{4-}$ polycrystals. **k** Detailed schematic illustration of the driving forces for the chiral regulation of assembly units in the polycrystals. Assembly units are represented by blue rods. The enlarged picture shows the specific steric configuration caused by the interactions between chiral ligands extended out of the (002) planes of the assembly unit surfaces of $Cu_4I_4(L_1)_4^{4+}$ $Cu_8I_{12}^{4-}$ polycrystals. Interactions between chiral ligands were represented by green shadings.

by the Van der Waals forces among chiral ligands extended out of the (002) planes between two surfaces of the assembly units of $Cu_4I_4(L_1)_4{}^{4+}$ $Cu_8I_{12}{}^{4-}$ polycrystals (Fig. 3k).

Similarly, such BNCC process can be also observed in the formation of $Cu_5I_7(L_2)_2$ polycrystals with drop-shaped micromorphology (Supplementary Fig. 20a–d). Nonetheless, the hierarchical crystallization process of $Cu_5I_7(L_4)_2$ is unique in comparison to the other three polycrystals, which went through microstructures of conglobate particles, nanosheets, and helical nanobelts, respectively (see Supplementary Note 2).

The crystallization monitoring also revealed that different crystallization rate induced by ligands correspondingly influenced hierarchical features of the obtained polycrystals. The short alkyl chain ligands would lead to the more rapid formation of nanoparticles than that for ligands with long chains. To be specific, PXRD patterns indicated that the crystalline phases of $Cu_4I_4(L_1)_4{}^{4+}$ $Cu_8I_{12}{}^{4-}$ and $Cu_5I_7(L_2)_2$ have already formed at 20 and 3 min, respectively. In contrast, the crystalline phase of $Cu_5I_7(L_3)_2$ did not form entirely in the initial reaction of 2 h. For $Cu_5I_7(L_4)_2$, the final nanobelts need a longer time (~7–12 h) to crystallize (Supplementary Figs. 18d and 20g, h). These are the fact that short alkyl chains absorbed on the assembly units lead to less repulsive forces than long alkyl chains (Supplementary Fig. 22). For the same reason, ligands with short terminal chains, such as $L_1$ and $L_2$, tend to induce assembly units to arrange more compactly (Fig. 3e and Supplementary Fig. 20d). Oppositely, $L_3$ and $L_4$ prefer to induce assembly in a sparser way derive from the relative independence of assembly units (Fig. 3i and Supplementary Fig. 20h).

**Hierarchical structure amplified chiroptical properties of polycrystals.** We measured the angle-dependent solid-state CD spectra of the obtained polycrystals via diffuse reflectance method to compare with that of single-crystal powders without hierarchical structure features (Supplementary Fig. 16 and Supplementary Fig. 23). The results suggested that CD signals displayed an order of magnitude improvement by this BNCC process, demonstrating the effective magnification of chirality via the hierarchical organization of chiral clusters. In addition, the CD signals of the obtained polycrystals are quite different from the corresponding CD signals of ligands, which indicates that the chirality of polycrystals is not originated from residual chiral ligands. Furthermore, we prepared polycrystals based on the racemic $L_1$ ligands through the same BNCC process. The obtained polycrystals present bundle structures that consisted of gathered parallel nanorods (Supplementary Fig. 25a) rather than helical structures of polycrystals induced by homochiral ligands. No corresponding CD and CPL signals were obtained for these racemic polycrystals (Supplementary Fig. 25b–d), which confirmed that the chirality of hierarchical polycrystals is indeed derived from homochiral hybrid clusters, implying the importance of homochiral clusters for stereoselectivity of hierarchical crystallization.

To further amplify the chirality by structural regulation, we explored the relationship between hierarchical features and chiroptical properties through changing the concentration of chiral ligands in the reaction solutions. The polycrystals obtained at different concentrations of ligands are all crystalline phases (Supplementary Fig. 26). Figure 4a–p and Supplementary Fig. 27a–p provide SEM images with high and low magnification, respectively. For $Cu_4I_4(L_1)_4{}^{4+}$ $Cu_8I_{12}{}^{4-}$, when the concentration of $L_1$ is low, the obtained polycrystals are rod-like morphology (Fig. 4a and Supplementary Fig. 27a), showing weak signals of CD (80 mdeg) and CPL ($g_{lum} = 9.6 \times 10^{-3}$) (Supplementary Fig. 28). With the increase of the concentration of $L_1$, excess chiral ligands absorbed on nanoparticle surfaces and existed in the solution make nanoparticles favor to attach mutually and twist via helical pattern,

forming the hierarchically structured chiral polycrystals (Fig. 4b and Supplementary Fig. 27b). The helical degree is maximum when the concentration of $L_1$ reaches to 6 mM (Fig. 4c and Supplementary Fig. 27c). Simultaneously, at this concentration, the chiroptical properties of the obtained polycrystals are strongest correspondingly. CD spectrum shows a positive Cotton effect and the CD signal reaches to 260 mdeg. The CPL signal is also strongest with the highest $g_{lum}$ value of $1.5 \times 10^{-2}$ (Supplementary Fig. 28). However, a greater amount of $L_1$ did not lead to a stronger chiral bias. On the contrary, when the concentration of $L_1$ is >7 mM, weaker CD signals and CPL emissions are observed. This is understandable because the reaction rate is accelerated with the increase of the $L_1$ concentration so that the initial nucleation and particle aggregates are completed more rapidly than the process of chiral regulation. Thus, a plenty of monomer particles aggregated more closely and attempted to form radially oriented nanorods converging to the center point without chiral structural features (Fig. 4d and Supplementary Fig. 27d).

For $Cu_5I_7(L_2)_2$ polycrystals, the bent garlic clove-liked shapes were observed when the concentration of $L_2$ is low (Fig. 4e and Supplementary Fig. 27e). These bent hierarchical structures do not exhibit strong CD and CPL due to their bent directions being random when they are dispersed in a solvent, leading to the same contribution of left-handed and right-handed CD or CPL signals (Supplementary Fig. 29). As the concentration of $L_2$ increased, the degree of bending reduced and polycrystals tended to display drop-liked shapes (Fig. 4f, g, and Supplementary Fig. 27f, g). Meanwhile, CD spectra showed positive Cotton effects and these drop-liked polycrystals exhibited the strongest CD and CPL with a $g_{lum}$ value of $4.0 \times 10^{-3}$ (Supplementary Fig. 29). When the concentration of $L_2$ increased to 30 mM, nanoparticles aggregated into glomerate structures, assimilating with that of $Cu_4I_4(L_1)_4{}^{4+}$ $Cu_8I_{12}{}^{4-}$ (Fig. 4h and Supplementary Fig. 27h), which presented weak CD and CPL (Supplementary Fig. 29).

For $Cu_5I_7(L_3)_2$ polycrystals, long lamellar units twined mutually into helically structured micromorphology. We found that helical structures always existed regardless of the concentration of $L_3$. When the concentration of $L_3$ was low, plenty of lamellar units twined densely and terminals of unit stretched out of the side so that the chiral scattering on the helical structure is not very obvious (Fig. 4i and Supplementary Fig. 27i), resulting in the relatively weak CD and CPL (Supplementary Fig. 30). However, the higher concentration of $L_3$ caused sparser units to twine together (Fig. 4j, k and Supplementary Fig. 27j, k), which is because the higher density of ligands on surfaces of lamellar units induced the helical assembly. Furthermore, with the increase of the concentration of $L_3$, the helicity of polycrystals became stronger and then prolonged (Fig. 4l and Supplementary Fig. 27l). As a consequence, the variation tendency of CD and CPL are much the same as that of helicity change of hierarchical structures, both of which are enhanced with the increase of concentration of $L_3$ (Supplementary Fig. 30).

With respect to $Cu_5I_7(L_4)_2$ polycrystals, as the concentration of $L_4$ increased, the variation tendency of hierarchical structures was similar to the structure change with the prolongation of the reaction time, which experienced through the conglobate particles, nanosheets, and helical nanobelts (Fig. 4m–p, Supplementary Fig. 27m–p). The chiroptical properties of these polycrystals are not strictly related to the microstructure variation because of their different crystal phases. The nanosheets with no helical structures can yield stronger CPL with a $g_{lum}$ value of $6.0 \times 10^{-3}$ and weaker CD than helical nanobelts (Supplementary Fig. 31).

**Circularly polarized phosphor performance for the application in LED device.** As a result of our above exploratory experiments

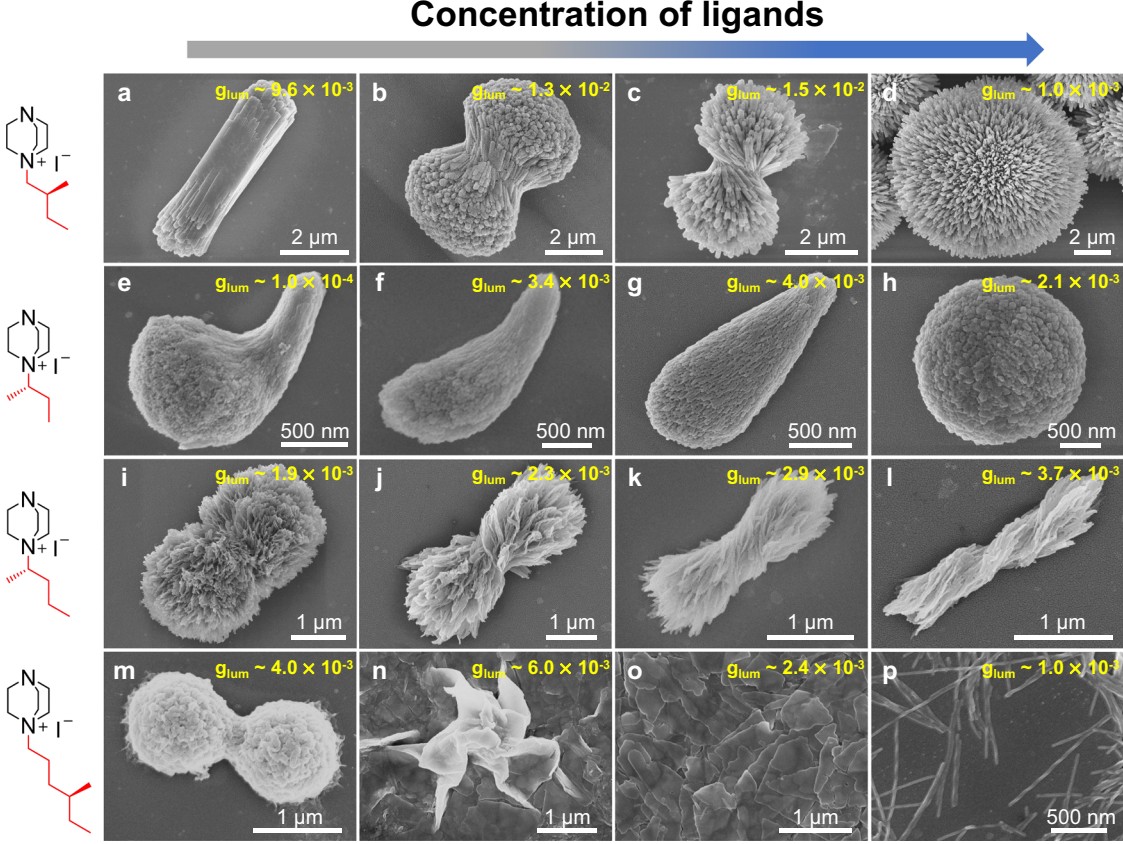

**Fig. 4 Influence of the concentration of ligands on the crystallization of hierarchically structured polycrystals. a–d** SEM images of $Cu_4I_4(L_1)_4^{4+}$ $Cu_8I_{12}^{4-}$ polycrystals with the concentrations of $L_1$ as 4 mM (**a**), 5 mM (**b**), 6 mM (**c**), 10 mM (**d**), respectively. **e–h** SEM images of $Cu_5I_7(L_2)_2$ polycrystals with the concentrations of $L_2$ as 5 mM (**e**), 10 mM (**f**), 20 mM (**g**), 30 mM (**h**), respectively. **i–l** SEM images of $Cu_5I_7(L_3)_2$ polycrystals with the concentrations of $L_3$ as 10 mM (**i**), 20 mM (**j**), 25 mM (**k**), 40 mM (**l**), respectively. **m–p** SEM images of $Cu_5I_7(L_4)_2$ polycrystals with the concentrations of $L_4$ as 3 mM (**m**), 5 mM (**n**), 10 mM (**o**), 20 mM (**p**), respectively.

on the regulation of ligand variety and concentration that play important roles in obtaining high-performance CPL-active polycrystals, we speculate the structure-property relationships for our hybrid cluster systems of BNCC. The luminescence dissymmetry factors of these polycrystals are highly related to the crystal morphology. Meanwhile, the twisted polycrystal morphology depends on the structure and the concentration of ligands. To be specific, on the one hand, the alkyl chain length may influence the crystal morphology and CPL performance. As shown in Fig. 5a, ligands with long alkyl chains at molecular terminals, such as $L_3$ and $L_4$, may cause low luminescent efficiency (PLQY ~ 7% and 6%), owing to the irregular vibration of alkyl chains in crystal lattices render the energy dissipation as heat. However, despite short alkyl chains contributing a lot to PLQY (50% for $Cu_5I_7(L_2)_2$), they may probably induce nanoparticles to aggregate more easily and quickly, leading to more compact aggregation, and then impede the formation of hierarchical structures without obvious chiral features (Fig. 4e–h). On the other hand, the position of the chiral center on the alkyl chain is another important factor for CPL performance. The chiral center away from the terminal of the alkyl chain, such as $L_2$ and $L_3$, may adversely impact on chiroptical properties owing to the inadequate interactions among chiral groups in the hierarchically structured polycrystals. As shown in Fig. 5b–d, the CD and CPL signals of $Cu_5I_7(L_2)_2$ and $Cu_5I_7(L_3)_2$ polycrystals are much weaker than that of $Cu_4I_4(L_1)_4^{4+}$ $Cu_8I_{12}^{4-}$ polycrystals. Taken together, among these ligands, $L_1$ possesses the proper chain length and position of chiral center, not only resulting in good luminescent

efficiency (PLQY ~ 32%) but also realizing effective chiral hierarchical crystallization at a certain concentration of ligands. As shown in Fig. 5d, the final optimized $Cu_4I_4(L_1)_4^{4+}$ $Cu_8I_{12}^{4-}$ chiral hierarchically structured polycrystals achieved the highest $g_{lum}$ value of $1.5 \times 10^{-2}$, which stands out among reported CPL materials (Supplementary Table 2).

These hierarchically structured chiral polycrystals present similar thermal stabilities to single crystals with the average thermal decomposition temperature > 200 °C (Supplementary Fig. 32), implying that they are suitable as phosphors for LED coating. To validate the feasibility of chiral polycrystals application in circularly polarized LED, we dispersed $Cu_4I_4(L_1)_4^{4+}$ $Cu_8I_{12}^{4-}$ polycrystals in the dichloromethane ($CH_2Cl_2$) solution of polymethyl methacrylate (PMMA) and coated them on the LED lamp. A uniform composite film of PMMA/chiral polycrystals was formed after solidification when $CH_2Cl_2$ evaporated. As shown in Fig. 5e, the fabricated LED device successfully realized bright green light emission with the chromaticity coordinate at (0.350, 0.571). We measured the circularly polarized light emitted by the fabricated LED device (details in Supplementary Fig. 33). There is a marked difference between the left- and right-handed light emission intensities, revealing the obvious circularly polarized light emitted by the fabricated LED device (Fig. 5f). The polarized degree (P) was calculated by

$$P = \frac{I_L - I_R}{I_L + I_R} \times 100\% \tag{2}$$

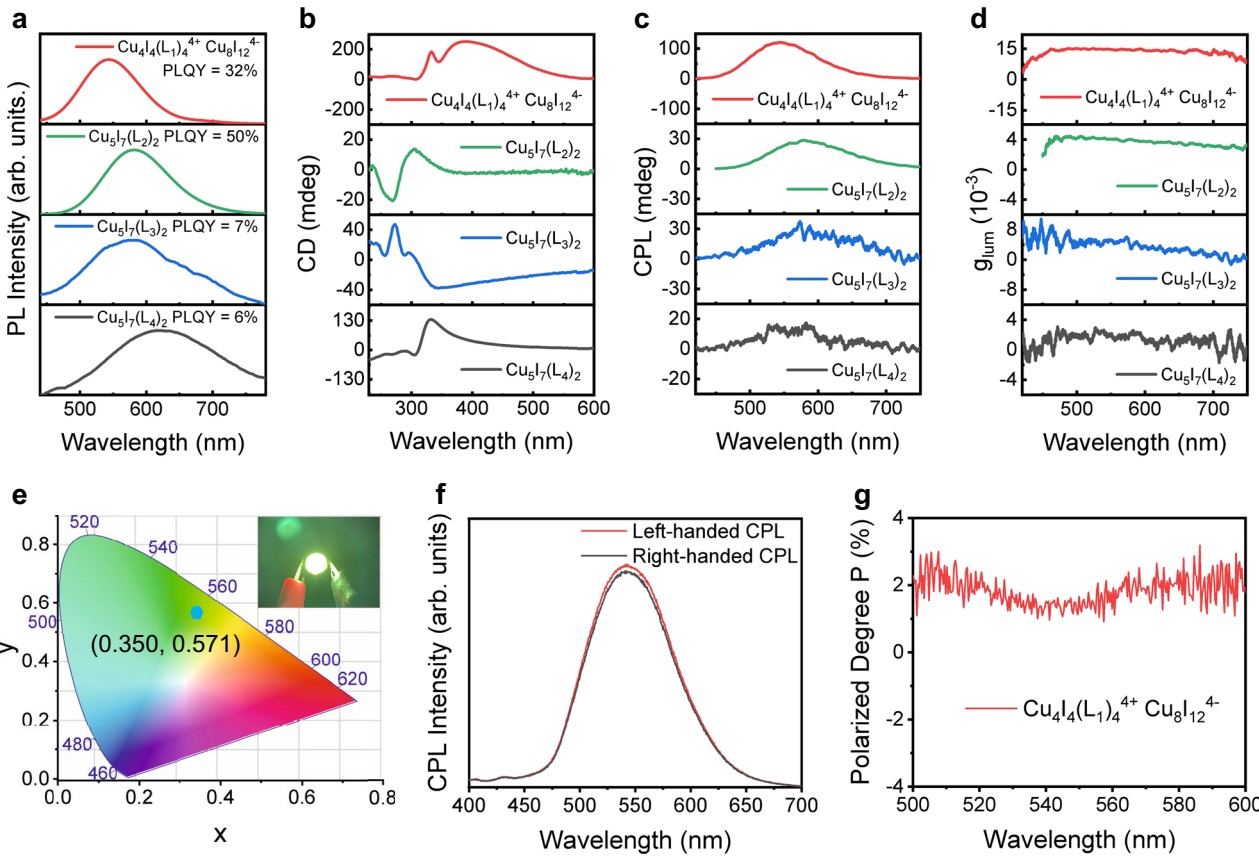

**Fig. 5 Chiroptical properties of hierarchically chiral polycrystals and circularly polarized characteristics of the LED device. a** PL spectra of $Cu_4I_4(L_1)_4^{4+}$ $Cu_8I_{12}^{4-}$, $Cu_5I_7(L_2)_2$, $Cu_5I_7(L_3)_2$, and $Cu_5I_7(L_4)_2$ polycrystals synthesized with different concentration of ligands: 6 mM ($L_1$), 20 mM ($L_2$), 40 mM ($L_3$), 20 mM ($L_4$). PLQY, the photoluminescence quantum yields. The PLQY of hierarchically structured polycrystals is 32%, 50%, 7%, 6% for $Cu_4I_4(L_1)_4^{4+}$ $Cu_8I_{12}^{4-}$, $Cu_5I_7(L_2)_2$, $Cu_5I_7(L_3)_2$ and $Cu_5I_7(L_4)_2$, respectively. **b-d** The corresponding CD spectra (**b**), CPL spectra (**c**), and $g_{lum}$ spectra (**d**). **e** Chromaticity coordinates graph of the LED device with the coating of $Cu_4I_4(L_1)_4^{4+}$ $Cu_8I_{12}^{4-}$ polycrystals. Inset: image of the fabricated LED device under operation. **f, g** CPL spectra (**f**) and the polarized degree analysis (**g**) of the fabricated LED device based on $Cu_4I_4(L_1)_4^{4+}$ $Cu_8I_{12}^{4-}$ polycrystals powered by 2.4 V. Source data are provided as a Source Data file.

Where $I_L$ and $I_R$ are emission intensities of left- and right-handed components of the circularly polarized LED device, respectively. The LED device exhibited a polarization degree of 1.84% in the wavelength range of 500–600 nm at room temperature (Fig. 5g), which successfully realized the demonstration of bright circularly polarized phosphor converted LED. In contrast, the LED device coated with racemic polycrystals did not exhibit any circularly polarized light emission (Supplementary Fig. 34).

In summary, we proposed a BNCC strategy to synthesize a series of hierarchically structured polycrystals with efficient CPL on the basis of rationally designed chiral clusters. We demonstrate that the obtained polycrystals can realize the magnification of chirality via the multiscale levels from molecular to nano- and then to mesoscopic scale. Furthermore, we unravel the non-classical crystallization of chiral clusters—mimicking biomineralization field—to demonstrate the formation process of these polycrystals that can be expanded to other chiral luminescent material systems. Eventually, a LED device with circularly polarized light emission of a polarization degree of 1.84% at room temperature was demonstrated, which provides a feasible strategy to fabricate an efficient circularly polarized light source.

## Methods

**Chemicals**. Cuprous iodide (CuI, 98%, Aladdin), potassium iodide (KI, 99%, Sinopharm Chemical Reagent Co. Ltd (SCRC)), Triethylenediamine (ted) (99%, Energy), (S)-2-methylbutan-1-ol (98%, TCI), (S)-pentan-2-ol (98%, TCI), (S)-butan-2-ol (98%, TCI), (S)-2-methylhexan-1-ol (98%, TCI), polyvinylpyrrolidone

(PVP K88-96, Mw = 1,300,000, Aladdin), iodine (99%, SCRC), triphenylphosphine (98%, SCRC), imidazole (99%, SCRC), dichloromethane (99%, SCRC), polymethyl methacrylate (PMMA, Macklin), ethanol (99.7%, SCRC), ethyl acetate (anhydrous, 99.8%, Alfa Aesar).

**Preparation of (S)-1-iodo-2-methylbutane**. Iodine (3.8 g, 15 mmol, 1.5 equiv.) was added into a solution of triphenylphosphine (3.93 g, 15 mmol, 1.5 equiv.) in $CH_2Cl_2$ (100 mL) at room temperature. The solution was allowed to stir for 10 min. Imidazole (1.7 g, 25 mmol, 2.5 equiv.) was subsequently added into the mixture above. After 10 min, chiral alcohol (S)-2-methylbutan-1-ol (0.88 g, 10 mmol, 1.0 equiv.) was dropped into the solution and the reaction system was stirred for 3 h. Then, the reaction mixture was quenched by the addition of the saturated aqueous solution of $Na_2S_2O_3$ (50 mL). Organic and aqueous layers were separated and the aqueous phase was extracted with $CH_2Cl_2$ (3 × 10 mL). The combined organic layers were dried with $Na_2SO_4$ (anhydrous). The mixture was filtered and the filtrate was concentrated to give a crude product. The residue was purified by silica gel column chromatography with hexane to get the colorless oily product (1.69 g, 85.3% yield).

**Preparation of (R)-2-iodobutane**. The synthetic procedure of (S)-2-iodobutane is similar to that of (S)-1-iodo-2-methylbutane. Only (S)-pentan-2-ol (0.74 g, 10 mmol, 1.0 equiv.) served as chiral alcohol. The colorless oily product was obtained (1.38 g, 75.0% yield).

**Preparation of (R)-2-iodoheptane**. The synthetic procedure of (R)-2-iodoheptane is similar to that of (S)-1-iodo-2-methylbutane. Only (S)-heptan-2-ol (0.88 g, 10 mmol, 1.0 equiv.) served as chiral alcohol. The colorless oily product was obtained (1.73 g, 87.3% yield).

**Preparation of (S)-1-iodo-4-methylhexane**. The synthetic procedure of (S)-iodo-4-methylhexane is similar to that of (S)-1-iodo-2-methylbutane. Only (S)-4-

methylhexan-1-ol (1.16 g, 10 mmol, 1.0 equiv.) served as chiral alcohol. The colorless oily product was obtained (2.20 g, 97.4% yield).

**Preparation of (S)-1-(2-methylbutyl)-1,4-diazabicyclo[2.2.2]octan-1-ium ($L_1$, (S)-2-Me-bu-ted) iodide**. (S)-1-iodo-2-methylbutane (1.60 g, 8 mmol, 1.0 equiv.) was added dropwise into ethyl acetate (60 mL) containing ted (1.35 g, 12 mmol, 1.5 equiv.) under magnetic stirring. The mixture was allowed to stir at room temperature and white solid precipitated out in 1 day. After this time, the generated precipitates were collected by centrifuged, washed with ethyl acetate, and dried under vacuum to give the pure product (1.56 g, 62.9% yield).

**Preparation of (S)-1-(sec-butyl)-1,4-diazabicyclo[2.2.2]octan-1-ium ($L_2$, (S)-1-Me-pr-ted) iodide**. $L_2$ was synthesized in a similar way to $L_1$. (R)-2-iodobutane served as chiral iodohydrocarbon (1.29 g, 7 mmol, 1.0 equiv.). The reaction mixture was stirred at room temperature for 3 days. The white solid product was obtained (1.91 g, 92.1% yield).

**Preparation of (S)-1-(pentan-2-yl)-1,4-diazabicyclo[2.2.2]octan-1-ium ($L_3$, (S)-1-Me-bu-ted) iodide**. $L_3$ was synthesized in a similar way to $L_1$. (R)-2-iodopentane served as chiral iodohydrocarbon (1.60 g, 8 mmol, 1.0 equiv.). The reaction mixture was stirred at room temperature for 3 days. The white solid product was obtained (1.51 g, 60.9% yield).

**Preparation of (S)-1-(4-methylhexyl)-1,4-diazabicyclo[2.2.2]octan-1-ium ($L_4$, (S)-4-Me-hex-ted) iodide**. $L_4$ was synthesized in a similar way to $L_1$. Only (S)-1-iodo-4-methylhexane served as chiral iodohydrocarbon (2.03 g, 9 mmol, 1.0 equiv.). The white solid product was obtained (2.74 g, 90.0% yield).

**Growth of single crystals of chiral hybrid clusters**. In a 5 mL vial, CuI (190 mg, 1 mmol) was dissolved in KI saturated solution (2 mL). Acetonitrile (1 mL) was added slowly along the inner wall into the bottom layer. Then, 0.5 M ethanol solution of $L_1$ (0.8 mL) (2 mL for $L_2$–$L_4$) was added slowly along the inner wall into the vial. Transparent single crystals formed overnight and were collected by filtration.

**Synthesis of hierarchically structured polycrystals**. In a 30 mL vial, 200 mg of polyvinylpyrrolidone (PVP K88-96) was added into 20 mL of ethanol under vigorous magnetic stirring. After PVP K88-96 was fully dissolved, KI saturated solution (0.4 mL) containing CuI (38 mg, 0.2 mmol) was added into the PVP/ethanol solution above. Then, a certain volume of ligands/ethanol solution (0.5 M) was added in. The mixture was allowed to vigorously stir for 12 h and then left standing overnight. The resulting white precipitation was collected by centrifugation at 6000 r/min for 3 min, washed with deionized water and ethanol several times via centrifugation. The obtained polycrystal powders were dispersed in ethanol for storing.

**The circularly polarized phosphor coating on the LED lamp**. In a 5 mL vial, 1 g of PMMA was dissolved in 2 mL of $CH_2Cl_2$. Then, the fabricated $Cu_4I_4(L_1)_4^{4+}$ $(Cu_8I_{12})^{4-}$ polycrystal powders with the strongest CPL signals were dispersed in PMMA/$CH_2Cl_2$ solution. The $Cu_4I_4(L_1)_4^{4+}$ $(Cu_8I_{12})^{4-}$ polycrystals dispersion was coated on an ultraviolet LED (3.6 V, 3.0 W, 365 nm) lamp bead. Solidification occurs when the $CH_2Cl_2$ evaporated, which gives a uniform film coating.

**Characterizations**. NMR spectra were recorded on Bruker 400 MHz spectrometers. The high-resolution mass spectra (HRMS) were recorded on Waters XEVO G2 Q-TOF (Waters Corporation). Optical rotations were measured by a Perkin-Elmer 343 polarimeter. Single-crystal X-ray diffraction (SCXRD) was performed on a Rigaku Oxford diffraction Gemini S Ultra diffractometer using Cu Kα radiation for $Cu_4I_4(L_1)_4^{4+}$ $(Cu_8I_{12})^{4-}$, and using Mo Kα radiation for $Cu_5I_7(L_2)_2$, $Cu_5I_7(L_3)_2$, and $Cu_5I_7(L_4)_2$. All single crystals were kept at 100 K during data collection. The data collection and processing were carried out with CrysAlisPro software. Using Olex2[36], the structures were solved with the ShelXT[37] structure solution program using Direct Methods and then refined with the ShelXL[38] refinement package using Least Squares minimization. Melting points were determined by a SWG X-4 melting apparatus. Fluorescence microscopy images were obtained by an OLYMPUS DP72 fluorescence microscopy using the WU filter with an X-Cite 120Q excitation light source system. UV–vis absorption spectra were recorded by a Varian Cary 300 UV–vis absorption spectrophotometer. Powder X-ray diffraction (PXRD) patterns were recorded using a Philips X'Pert PRO SUPER X-ray diffractometer with Cu Kα radiation. Photoluminescence (PL) spectra and photoluminescence quantum yields (PLQY) were collected by a Hamamatsu absolute PL yield spectrometer C11347, using 350-nm excitation on powders or ethanol suspensions in a quartz cuvette at room temperature. Thermogravimetric analyses (TGA) curves were recorded by a Q5000IR thermal gravimetric analyzer with a heating rate of 10 °C/min under nitrogen atmosphere. PL lifetime measurements were carried out on a Deltaflex Steady-state/Lifetime Spectrofluorometer. Scanning electron microscopy (SEM) images were taken by A Carl Zeiss Supra 40 field emission scanning electron microscope with an accelerating voltage of 5 kV. All samples were observed after gold sputtering for 60 s with a constant current of

30 mA. Transmission electron microscopy (TEM) images were taken by a Hitachi HT-7700 transmission electron microscope with an accelerating voltage of 120 kV. The high-resolution transmission electron microscopy (HRTEM) images and the selected area electron diffraction (SAED) were acquired on a Talos F200X high-resolution transmission electron microscopy. Circular dichroism (CD) spectra were recorded by JASCO-1700 or JASCO-1500 Circular Dichroism Spectrophotometer using diffuse reflectance method for solid-state samples and a JASCO-810 Circular Dichroism Spectrophotometer using transmission method for the ethanol dispersion of polycrystals with "standard" sensitivity at 100 nm min$^{-1}$ scan speed. The final solid-state CD signals are the average value of each rotation of 45 degrees to eliminate the influence of linear dichroism. Circularly polarized luminescence (CPL) spectra were recorded on a JASCO CPL-300 Spectrophotometer with excitation of 350 nm at a scan speed of 100 nm min$^{-1}$. The chromaticity coordinate of coated LED was measured by PR-670 spectrometer. The spectra of left- and right-handed circularly polarized light of fabricated LED device were recorded by FLMS12313 spectrometer.

## Data availability

The data supporting the findings of this study are available within the Article and its Supplementary Information. Source data are provided with this paper. The crystal structures in this work were deposited in Cambridge Crystallographic Data Centre (CCDC) with the deposition numbers 2125284, 2125285, 2125286, and 2125291 for $Cu_4I_4(L_1)_4^{4+}$ $Cu_8I_{12}^{4-}$, $Cu_5I_7(L_2)_2$, $Cu_5I_7(L_3)_2$, and $Cu_5I_7(L_4)_2$, respectively. These data can be obtained free of charge from the CCDC via http://www.ccdc.cam.ac.uk/data_request/cif. Source data are provided with this paper.

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

## Acknowledgements

We acknowledge the financial support from the National Natural Science Foundation of China (Grant No. 52073271, 21875236, 21573211, 21633007 and 22161142004), the Fundamental Research Funds for the Central Universities (Grant No. WK2060190085), the Joint Funds from Hefei National Synchrotron Radiation Laboratory (Grant No. KY2060000172) and the State Key Laboratory of Luminescence and Applications (Grant SKLA-2020-06). We also thank the support from the USTC Center for Micro and Nanoscale Research and Fabrication.

## Author contributions

H.-B.Y. conceived the idea, designed the experiment and analyzed the data. L.-Z.F. synthesized the materials, carried out characterizations and analyzed the data. J.-J.W. participated in data analyses and conducted the TEM characterization. K.-H.S. and T.Z. participated in data analyses and manuscript organization. T.M. and Y.-C.Y. drew schematic diagram of assembly and crystallization process. Z.-D.L. and F.-J.F. provided assistance for emission polarization measurements of LED. M.-M.Z., S.J., and M.-Z.Z. provided the supports for CPL measurements. L.-Z.F. and H.-B.Y. co-wrote the manuscript. H.-B.Y. directed and supervised the project. All authors contributed to discussions and finalizing the manuscript.

## Competing interests

The authors declare no competing interests.
