## [Peer review file · Nature Communications]

REVIEWER COMMENTS

Reviewer #1 (Remarks to the Author):

The authors designed several kinds of novel copper(I)-iodide clusters hybridized with chiral ligands in this manuscript. They first resolved the crystal structures of the new materials, and then further tailored the microscopic morphologies of the materials to enhance the luminescent efficiency. The formation of the hierarchical structures was attributed to a biomimetic non-classical crystallization (BNCC) process, which includes a nucleation process, a particle-aggregation and oriented-attachment process, and a mesoscopic transformation process. The relationship between morphologies and luminescence dissymmetry factors was also investigated. Then the authors explored the circularly polarized phosphor performance of the materials for the application in LED devices, and obtained an impressive polarization degree compared with other reported results. Inorganic chirality is now a burgeoning hot topic due to its versatility and good stability, which might hold promise for new technologies and smart optical-based devices. However, designing purely inorganic functional chiral systems is challenging. Fabrication of inorganic-organic hybrid functional chiral systems might be another choice. The reported materials here showed good stability like inorganics. In addition, the non-classical crystallization mechanism could also shed light on designing chiral inorganic materials. The results here are important for both the fields of material fabrications and circularly polarized phosphor, and would surely attract wide attention in the relevant fields. Thus, I would recommend the publication of this manuscript after addressing my following concerns.

1. The helical arrangements of clusters in the lattices are not so obvious, especially in the cases of $\text{Cu}_5\text{I}_7(\text{L}_3)_2$ and $\text{Cu}_5\text{I}_7(\text{L}_4)_2$. It seems that these arrangements appear to be helical when virtual cylinders are placed anywhere in the crystal structure. The authors need to give more attention to this statement.
2. The BNCC process described in the manuscript remains superficial. First of all, which biological case does this non-classical crystallization process mimic? Particle aggregations and oriented attachments are also widespread in inorganic minerals and nanoparticle synthesis. The twisted shapes are not a structure unique to biological materials. Moreover, the authors could only obtain twisted shapes under certain conditions. Secondly, the structures of particles at each step were not adequately characterized, and the driven forces for the movement of particles to the next stage are not researched. They summarized the mechanism based on the SEM images obtained at different stages, however, no in-depth study was carried out. It is recommended to perform SAEDs on particles at each step to check the crystallinity changes. This might help to understand whether the formation of nanocrystals at the initial stages is classical or non-classical, as well as the evolution of crystallinities of the single-particle. It is also recommended that zeta potential measurements are used to gain insight into the driving forces.
3. To which particle and to what stage does the lattice in Fig. 3j belong? Furthermore, the facet analysis according to the SAED (Fig. 3j, inset) seemed to contradict the lattice fringe analysis. And these results cannot reveal the exposed facets since the shape of the particle appears to be irregular. Nor do these results imply a driving force for the assembly.

4. The luminescence dissymmetry factors of materials shown in Fig. 4 were related to both the material morphology and chiral ligands. Therefore, it's difficult to gain insights into the chemical design of chiral ligands to enhance the dissymmetry factors. Can the authors establish a structure-property relationship based on the chemical structures?

5. In the ligand concentration studies, were the contents of chiral ligands in the final materials equal? Is it possible that the difference in g factors originated from the residual chiral ligands?

6. The authors have misused the term "mesocrystal" in some places. The term was coined to define assembling structures with similar lattice orientations. However, the authors didn't provide any evidence on the lattice orientations.

Bing Ni

University of Konstanz

Reviewer #2 (Remarks to the Author):

This manuscript demonstrated the method of biomimetic non-classical crystallization strategy to prepare kinds of hierarchically structured polycrystals materials with circularly polarized luminescence (CPL) activity. The crystallization process was well observed, and good chiroptical properties in the corresponding light emitting diode devices are achieved. However, to fully reveal the mechanism and relative properties, corresponding data are insufficient to support. After careful reading of this manuscript, I think it should be published after dealing with the following questions:

1. Page 3, Line 83, the sentence "However, the well-developed hierarchical construction..." is not suitable to describe the research gap about this work, because this manuscript is still based on the strategy of chiral assembly to amplify CPL and enhancing quantum efficiency by increasing molecular rigidity. It is similar to the method to obtain CPL from chiral aggregation-induced emission (AIE) molecules and macrostructures, and high-quality CPL also could be observed, such as *J. Mater. Chem. C*, 2015,3, 6997-7003. *Mater. Horiz.*, 2020,7, 3209-3216. As compared to this strategy, please supplement the unique advantages and innovation in the introduction part.

2. Page 11, Line 261, Authors regard the weaker CD and CPL signals as the result of accelerated crystallization processes. Can it be understood that excellent chiral structure in this system is formed by a thermodynamic process with more regular crystallization? It is supposed to compare the crystallinity at different concentrations.

3. In all CD and CPL measurements of chiral samples, this manuscript lacks data of enantiomeric related properties.

4. Considering the pseudo signal of linear polarization for crystal structure, angle-dependent CD and CPL tests should be given.

Reviewer #3 (Remarks to the Author):

In this manuscript, the authors report an interesting biomimetic non-classical crystallization (BNCC) strategy to construct a series of hierarchically structured chiral Cu-I clusters, which exhibit excellent CPL performance. The authors first rationally designed four chiral ligands to build up robust chiral Cu-I-ligand hybrid clusters. Using these unique and robust hybrid clusters, the authors developed an interesting BNCC process to fabricate hierarchically structured CPL-phosphors, showing an impressive concept in the field of CPL-active cluster based materials. More impressively, the obtained CPL-active phosphors exhibited excellent CPL with high PLQYs (32%) and high glum value (1.5×10^{-2}), which enabled a circularly polarized phosphor converted LED with a polarization degree of 1.84%. These new findings represent synthetic strategy and BNCC breakthroughs for synthesizing high performance CPL-active phosphors and fabricating circularly polarized LEDs. Therefore, I strongly recommend the publication of this manuscript in Nature Communications. The following minor comments are just provided to further strength this work before its publication.

1. The authors should present the certain helical arrangements orientations (P/M) of crystal structures for clarity to specify the detailed chiral information in the obtained phosphors.

2. In order to explain the transfer of chirality from ligand to the cluster and then to the phosphor powder, the CD spectra of ligands should be provided to compare with single crystal powders and polycrystals.

3. It is very interesting to see that different ligand and cluster design results in different performance in CPL such as photoluminescence and glum. If the authors can provide some summary and discussions on this structural design-optical performance correlations, that will further strength the mechanism illustration in this work.

4. The CCDC deposition numbers of all crystals should be listed in the manuscript for the sake of regulation.

5. The word "well-defined" at line 98, page 4 should be "well-designed"; the words "was still existed" in line 201, page 8 should be "still exist"; the word "synthesized" in line 338, page 15 should be "synthesize".

The point-by-point response to the reviewers' comments

We have carefully considered the valuable comments and suggestions from 3 reviewers, and have addressed them here and made revisions in the manuscript accordingly.

Point-by-point response to reviewer #1's comments

Overall Comments:

“The authors designed several kinds of novel copper(I)-iodide clusters hybridized with chiral ligands in this manuscript. They first resolved the crystal structures of the new materials, and then further tailored the microscopic morphologies of the materials to enhance the luminescent efficiency. The formation of the hierarchical structures was attributed to a biomimetic non-classical crystallization (BNCC) process, which includes a nucleation process, a particle-aggregation and oriented-attachment process, and a mesoscopic transformation process. The relationship between morphologies and luminescence dissymmetry factors was also investigated. Then the authors explored the circularly polarized phosphor performance of the materials for the application in LED devices and obtained an impressive polarization degree compared with other reported results. Inorganic chirality is now a burgeoning hot topic due to its versatility and good stability, which might hold promise for new technologies and smart optical-based devices. However, designing purely inorganic functional chiral systems is challenging. Fabrication of inorganic-organic hybrid functional chiral systems might be another choice. The reported materials here showed good stability like inorganics. In addition, the non-classical crystallization mechanism could also shed light on designing chiral inorganic materials. The results here are important for both the fields of material fabrications and circularly polarized phosphor, and would surely attract wide attention in the relevant fields. Thus, I would recommend the publication of this manuscript after addressing my following concerns.”

Reply: We appreciate the reviewer's highly positive comment. We tried our best effort to address all the detailed comments and made a detailed revision of our manuscript.

Comment 1): *“The helical arrangements of clusters in the lattices are not so obvious, especially in the cases of $Cu_5I_7(L_3)_2$ and $Cu_5I_7(L_4)_2$. It seems that these arrangements*

appear to be helical when virtual cylinders are placed anywhere in the crystal structure. The authors need to give more attention to this statement.”

Reply: We thank the reviewer for the suggestive comments. We are sorry for our ambiguous descriptions of spatial helical arrangements of the hybrid clusters in crystal structures. The revised spatial arrangements of the clusters in the lattices along different directions are shown in Replied Fig. 1-4. After our careful study, we confirmed the symmetry elements in the obtained four crystals. To be specific, 4-fold and 3-fold screw axes can be found in $\text{Cu}_4\text{I}_4(\text{L}_1)_4^{4+} \text{Cu}_8\text{I}_{12}^{4+}$ and $\text{Cu}_5\text{I}_7(\text{L}_3)_2$ crystals, which is due to the $P4_2$ and $P3_221$ space groups (Replied Fig. 5-6). However, there are no symmetry elements for $\text{Cu}_5\text{I}_7(\text{L}_2)_2$ and $\text{Cu}_5\text{I}_7(\text{L}_4)_2$ crystals because of the $P1$ space group, which has the lowest symmetry degree in the space groups. *We have corrected the presentation of Fig. 1 and have added the statements and presentations of cluster arrangements of four crystals in the revised manuscript (Page 6, line 106, 113) and Supporting Information (Supplementary Fig. 11-16).*

Replied Fig. 1. The spatial arrangement of $\text{Cu}_4\text{I}_4(\text{L}_1)_4^{4+} \text{Cu}_8\text{I}_{12}^{4-}$ hybrid clusters in crystal lattices along a, b, c axes. Color scheme: Cu, red; I, yellow; N, blue; C, Gray.

Replied Fig. 2. The spatial arrangement of $\text{Cu}_5\text{I}_7(\text{L}_2)_2$ hybrid clusters in crystal lattices along a, b, c axes. Color scheme: Cu, red; I, yellow; N, blue; C, Gray.

Replied Fig. 3. The spatial arrangement of $\text{Cu}_5\text{I}_7(\text{L}_3)_2$ hybrid clusters in crystal lattices along a, b, c axes. Color scheme: Cu, red; I, yellow; N, blue; C, Gray.

Replied Fig. 4. The spatial arrangement of $\text{Cu}_5\text{I}_7(\text{L}_4)_2$ hybrid clusters in crystal lattices along a, b, c axes. Color scheme: Cu, red; I, yellow; N, blue; C, Gray.

Replied Fig. 5. 4-Fold screw axis of $\text{Cu}_4\text{I}_4(\text{L}_1)_4^{4+} \text{Cu}_8\text{I}_{12}^{4-}$ hybrid clusters in crystal lattices. Color scheme: Cu, red; I, yellow; N, blue; C, Gray.

Replied Fig. 6. 3-Fold screw axis of $\text{Cu}_5\text{I}_7(\text{L}_4)_2$ hybrid clusters in crystal lattices. Color scheme: Cu, red; I, yellow; N, blue; C, Gray.

Comment 2): “The BNCC process described in the manuscript remains superficial. First of all, which biological case does this non-classical crystallization process mimic? Particle aggregations and oriented attachments are also widespread in inorganic minerals and nanoparticle synthesis. The twisted shapes are not a structure unique to biological materials. Moreover, the authors could only obtain twisted shapes under certain conditions. Secondly, the structures of particles at each step were not adequately characterized, and the driven forces for the movement of particles to the next stage are not researched. They summarized the mechanism based on the SEM images obtained at different stages, however, no in-depth study was carried out. It is recommended to perform SAEDs on particles at each step to check

the crystallinity changes. This might help to understand whether the formation of nanocrystals at the initial stages is classical or non-classical, as well as the evolution of crystallinities of the single-particle. It is also recommended that zeta potential measurements are used to gain insight into the driving forces.”

Reply: We thank the reviewer for the valuable comments and suggestions. It will be very interesting to find out a specific biological case that the non-classical crystallization process to mimic and this is also very challenging. In our work, the proposed BNCC process here does not correlate to a specific biological case, which is a general pathway of biomineralization in biological materials. The non-classical crystallization process is widely found in many biominerals, such as sea urchin spikes (Oaki, Y. & Imai, H. *Small* 2, 66-70 (2006)) and aragonite tablets in nacre (Oaki, K. & Imai, H. *Angew. Chem. Int. Ed.* 44, 6571-6575 (2005); Rousseau, M. et al. *Biomaterials* 26, 6254-6262 (2005)). In these constructions of superstructures in biominerals, inorganic units usually coordinate exquisitely with organic molecules and polymers into hierarchically structured assemblies. Therefore, here, a similar strategy is extended to realize the construction of hierarchically structured chiral luminescent materials to amplify CPL, which can be regarded as generally mimicking the formation of the hierarchical biomineral.

To adequately characterize the structure of particles at each step and confirm the non-classical crystallization of the particles at initial stage, we supplement TEM and corresponding SAEDs analysis at different crystallization stages (Replied Fig. 7). The TEM images display the morphological change from nanoparticles to aggregates and then to hierarchically structured intermediates and to finally crystallized hierarchical structure composed of nanorods (Replied Fig. 7a-d). This variation revealed by TEM is similar to the SEM results but could provide more information about the crystallization process. Specifically, the corresponding SAEDs patterns clearly show the variation of the obtained diffraction patterns from diffraction rings to clear diffraction spots, which indicates the non-classical crystallization process of our hybrid chiral cluster system. At the first stage, the initially formed nanoparticles present the amorphous state (Replied Fig. 7e). Then, these amorphous nanoparticles aggregated into microparticles under the influence of ligands and polymers still with low crystallinity (Replied Fig. 7f). With prolonging the processing time, the aggregated microparticles evolved into denser structure with improve polycrystalline characteristics (Replied Fig. 7g). Finally, the highly crystalline hierarchical structures with crystalline nanorods inside are obtained (Replied Fig. 7h). This evolution of crystallinity of single particle in the hierarchical structures clearly confirmed the non-classical crystallization process of our hybrid chiral clusters.

In addition, zeta potentials of reaction system of $\text{Cu}_4\text{I}_4(\text{L}_1)_4^{4+}$ $\text{Cu}_8\text{I}_{12}^{4-}$ polycrystals at different reaction time are relatively low (-11 mV ~ -2 mV) compared to typical

charged colloidal nanocrystals (< -30 mV or > 30 mV) (Replied Fig. 8). This is probably because the neutral polymer PVP only has permanent dipoles which originate from polar C=O bonds, rather than free charges. Thus, in our case, the electrostatic repulsion is insufficient to stabilize the formed colloidal systems and the nanoparticles tend to aggregate and assembly through the driving force of lattice formation energy of the hybrid cluster crystals. From the above analysis, the steric hindrance from the entanglement among PVP chains is the primary stable factor in the initial reaction period. As the reaction progresses, the nanoparticles collide with each other. And the lattice formation energy and dipolar-dipolar interactions acting as the driving forces induce the transformation of particles into crystalline hierarchical structures. Meanwhile, the Van der Waals force among the chiral ligands is the primary factor of the chiral regulation instead of driving forces for the assembly.

We have added the TEM and SAEDs results in the Supporting Information (Supplementary Fig. 20) and made a more detailed explanation of the non-classical crystallization in the revised manuscript (Page 8, line 163; Page 9, line 178).

Replied Fig. 7. a-h, TEM images (**a-d**) and SAEDs patterns (**e-h**) at different stages of crystallization for $\text{Cu}_4\text{I}_4(\text{L}_1)_4^{4+} \text{Cu}_8\text{I}_{12}^{4-}$ polycrystals with the reaction time of 3 min (**a, e**), 20 min (**b, f**), 1 h (**c, g**), 6 h (**d, h**), respectively ($[\text{L}_1] = 6$ mM).

Replied Fig. 8. Zeta potential curve of the colloidal particles at different reaction time of crystallization for $\text{Cu}_4\text{I}_4(\text{L}_1)_4^{4+} \text{Cu}_8\text{I}_{12}^{4-}$ polycrystals.

Comment 3): “To which particle and to what stage does the lattice in Fig. 3j belong? Furthermore, the facet analysis according to the SAED (Fig. 3j, inset) seemed to contradict the lattice fringe analysis. And these results cannot reveal the exposed facets since the shape of the particle appears to be irregular. Nor do these results imply a driving force for the assembly.”

Reply: We thank the reviewer for the careful check. The lattice in Fig. 3j belongs to the $\text{Cu}_4\text{I}_4(\text{L}_1)_4^{4+} \text{Cu}_8\text{I}_{12}^{4-}$ polycrystals at the final crystallization stage. The lattice fringes of HRTEM and SAEDs are different because the capture area of HRTEM is different from that of SAEDs because we carry out the characterization on different samples. To avoid this misleading result, we removed the SAED pattern in the Fig.3j in the revised manuscript. As the reviewer pointed out that the exposed facets cannot be determined in our TEM characterization, we here think that Van der Waals force between chiral alkyl chains extended out of exposed crystal planes is one of the driving forces for the chiral regulation (Fig. 3k in main text) rather than assembly for crystallization.

Comment 4): “The luminescence dissymmetry factors of materials shown in Fig. 4 were related to both the material morphology and chiral ligands. Therefore, it’s difficult to gain insights into the chemical design of chiral ligands to enhance the dissymmetry factors. Can the authors establish a structure-property relationship based on the chemical structures?”

Reply: We thank the reviewer for this constructive comment. The luminescence dissymmetry factors of these polycrystals are highly related to the hierarchical morphology for CPL amplification. Meanwhile, the twisted crystal morphology depends on the crystal structure and the concentration of ligands in the reaction system. Thus, the structure-property relationship based on the chemical structures can be summarized as followings. Firstly, at the stage of nanoparticle aggregation, the length of the alkyl chain of the ligand is the most important factor in the initial shape of polycrystals. Specifically, the shorter alkyl chain may induce nanoparticles to aggregate more easily and quickly, which will induce nanoparticles to gather compactly. On the contrary, the longer alkyl chain will retard the aggregation so that there is more time to execute the chiral regulation. So the nanoparticles tend to aggregate into a specific shape with chirality. Secondly, at the stage of oriented attachment, the position of the chiral center on the alkyl chain is also an important factor in the finally obtained hierarchical morphology. If the position of the chiral center is away from the terminals of alkyl chains, such as L_2 and L_3 , the chiral regulation between crystal boundaries will be inadequate because of the lack of interactions. In addition, the luminescent efficiency is also affected by the chain length of the ligand. A longer alkyl chain will give rise to a lower PLQY. Thirdly, the chiroptical properties are also influenced by the concentration of ligands in the reaction system. A small or excessive amount of ligands in the reaction system are both detrimental to the chiroptical properties of the finally obtained polycrystals because helical morphologies cannot be obtained in these cases. In summary, to obtain the high performance CPL materials with both high luminescent efficiency and dissymmetry, the chemical designed principle of ligands should be: (1) an appropriate chain length; (2) the position of the chiral center located near the terminal of ligand chain; (3) an appropriate ligand concentration in the reaction system. *We have added detailed discussion about this issue in our revised manuscript (Page 13, line 275-292).*

Comment 5): “In the ligand concentration studies, were the contents of chiral ligands in the final materials equal? Is it possible that the difference in g factors originated from the residual chiral ligands?”

Reply: We thank the reviewer for the thoughtful comments. For the finally obtained hierarchical polycrystals, we washed them with deionized water and ethanol several times via centrifugation. So we think that the amount of residual chiral ligands in the obtained polycrystals is very tiny and the influence of residual chiral ligands on the g factors is weak. In addition, we supplemented the CD spectra of four ligands. As shown in Replied Fig. 9, CD signals of the chiral ligands are different from that of polycrystals, which means that the difference of g factors cannot be originated from residual chiral ligands. *We have added the CD spectra of ligands in the revised Supporting Information (Supplementary Fig. 25). We have also discussed the influence of residual ligands in the revised manuscript (Page 10, line 213).*

Replied Fig. 9. CD spectra of ligands L₁, L₂, L₃, and L₄.

Comment 6): “The authors have misused the term ‘mesocrystal’ in some places. The term was coined to define assembling structures with similar lattice orientations. However, the authors didn’t provide any evidence on the lattice orientations.”

Reply: We appreciate the reviewer for pointing out this misuse. We have corrected the ‘mesocrystal’ as the term “attached-nanocrystal” in the revised manuscript to express the crystal particles attached into a regular shape.

Point-by-point response to reviewer #2's comments

Overall Comments:

“This manuscript demonstrated the method of biomimetic non-classical crystallization strategy to prepare kinds of hierarchically structured polycrystals materials with circularly polarized luminescence (CPL) activity. The crystallization process was well observed, and good chiroptical properties in the corresponding light emitting diode devices are achieved. However, to fully reveal the mechanism and relative properties, corresponding data are insufficient to support. After careful reading of this manuscript, I think it should be published after dealing with the following questions:”

Reply: We appreciate the reviewer's nice comments. We tried our best effort to address all the detailed comments and made a detailed revision of our manuscript.

Comment 1): “Page 3, Line 83, the sentence “However, the well-developed hierarchical construction...” is not suitable to describe the research gap about this work, because this manuscript is still based on the strategy of chiral assembly to amplify CPL and enhance quantum efficiency by increasing molecular rigidity. It is similar to the method to obtain CPL from chiral aggregation-induced emission (AIE) molecules and macrostructures, and high-quality CPL also could be observed, such as J. Mater. Chem. C, 2015, 3, 6997-7003. Mater. Horiz., 2020,7, 3209-3216. As compared to this strategy, please supplement the unique advantages and innovation in the introduction part.”

Reply: We thank the reviewer for the thoughtful comments and valuable suggestions. We are sorry for our negligence of the significant progresses on the CPL materials based on AIE molecules and macrostructures. Our work about the Cu-I organic-inorganic hybrid material system and their assemblies to fabricate hierarchically structured chiral polycrystals guided by biomimetic non-classical crystallization is similar to AIE processed CPL-active materials, aiming to realize the balance between luminescent efficiency and g_{lum} of CPL materials. In comparison to the AIE CPL-active materials based on organic molecules, our hybrid material has the advantages of synthetic simplicity and good stability like the inorganic system as pointed out by the reviewer 1. Based on our well-designed Cu-I organic-inorganic hybrid clusters, we have also studied the structure-property relationship and proposed

a principle of chemical design of chiral ligands in the BNCC process to gain the optimal CPL-active materials. *In the revised manuscript, we tried our best to clearly acknowledge and thoroughly discuss previous works in the introduction part (Page 3, line 50).*

Comment 2): “Page 11, Line 261, Authors regard the weaker CD and CPL signals as the result of accelerated crystallization processes. Can it be understood that excellent chiral structure in this system is formed by a thermodynamic process with more regular crystallization? It is supposed to compare the crystallinity at different concentrations.”

Reply: We thank the reviewer for pointing out this issue. The crystallization process in our reported hierarchical hybrid cluster polycrystals is a non-classical crystallization process, which is a dynamic process not the thermodynamic process. In the case of $\text{Cu}_4\text{I}_4(\text{L}_1)_4^{4+} \text{Cu}_8\text{I}_{12}^{4-}$ hierarchical polycrystals with the weaker CD and CPL as result of increased L_1 concentration, our description about the accelerated crystallization process is ambiguous. In fact, the high concentration of L_1 resulted in the accelerated chemical reaction rate of hybrid clusters, leading to rapid initial nucleation and particle aggregates of our reported non-classical crystallization process with dynamic control not thermodynamic. The rapid initial nucleation and particle aggregates out of chiral ligand regulation finally resulted in the formation of polycrystals without chiral structure features, which display weaker CD and CPL signals. *We have modified the corresponding description in the revised manuscript (Page 12, line 239)*

Comment 3): “In all CD and CPL measurements of chiral samples, this manuscript lacks data of enantiomeric related properties.”

Reply: We thank the reviewer for this concern. In our reported hybrid clusters, four S-isomer chiral ligands were synthesized from commercialized S-isomer chiral alcohols, which can be extracted from natural products. However, it is very hard to get R-isomer chiral ligands via direct chemical synthesis without corresponding natural sources. Thus, in our work, the enantiomeric related properties of hybrid polycrystals based on R-isomer chiral ligands are lack.

Comment 4): “Considering the pseudo signal of linear polarization for crystal structure, angle-dependent CD and CPL tests should be given.”

Reply: We thank the reviewer for this constructive suggestion. Our CD and CPL tests were performed on fully ground crystal powders, in which the orientation of the crystals is random, and thus the pseudo signal of linear polarization for crystal structure is largely suppressed. To confirm this, we supplied the angle-dependent solid CD spectra for single crystal powders. As shown in Replied Fig. 10, the CD signals collected from different angles are similar without any significant pseudo signal of linear polarization, which indicates that CD and CPL tests on the fully ground crystal powders are angle-independent. Moreover, CD and CPL spectra of the hierarchically structured polycrystals were collected using the transmission method with randomly dispersed polycrystal powders in the ethanol. Therefore, the pseudo signal from linear polarization is also negligible for the CD and CPL tests based on highly dispersed hierarchically structured polycrystals in the ethanol. *We have added angle-dependent solid CD spectra of single crystal powders in the revised Supporting Information (Supplementary Fig. 17) and Methods (Page 19, line 408).*

Replied Fig. 10. Angle-dependent CD spectra of $\text{Cu}_4\text{I}_4(\text{L}_1)_4^{4+} \text{Cu}_8\text{I}_{12}^{4-}$, $\text{Cu}_5\text{I}_7(\text{L}_2)_2$, $\text{Cu}_5\text{I}_7(\text{L}_3)_2$, and $\text{Cu}_5\text{I}_7(\text{L}_4)_2$ fully ground single crystal powders.

Point-by-point response to Reviewer #3's comments

Overall Comments:

“In this manuscript, the authors report an interesting biomimetic non-classical crystallization (BNCC) strategy to construct a series of hierarchically structured chiral Cu-I clusters, which exhibit excellent CPL performance. The authors first rationally designed four chiral ligands to build up robust chiral Cu-I-ligand hybrid clusters. Using these unique and robust hybrid clusters, the authors developed an interesting BNCC process to fabricate hierarchically structured CPL-phosphors, showing an impressive concept in the field of CPL-active cluster based materials. More impressively, the obtained CPL-active phosphors exhibited excellent CPL with high PLQYs (32%) and high glum value (1.5×10^{-2}), which enabled a circularly polarized phosphor converted LED with a polarization degree of 1.84%. These new findings represent synthetic strategy and BNCC breakthroughs for synthesizing high performance CPL-active phosphors and fabricating circularly polarized LEDs. Therefore, I strongly recommend the publication of this manuscript in Nature Communications. The following minor comments are just provided to further strengthen this work before its publication.”

Reply: We appreciate the reviewer's highly positive comments.

***Comment 1):** “The authors should present the certain helical arrangements orientations (P/M) of crystal structures for clarity to specify the detailed chiral information in the obtained phosphors.”*

Reply: We thank the reviewer for this suggestion. After our careful study, we have redefined the spatial arrangements of the clusters in the crystal lattices. The space groups are $P4_2$, $P1$, $P3_221$, and $P1$ for $\text{Cu}_4\text{I}_4(\text{L}_1)_4^{4+} \text{Cu}_8\text{I}_{12}^{4-}$, $\text{Cu}_5\text{I}_7(\text{L}_2)_2$, $\text{Cu}_5\text{I}_7(\text{L}_3)_2$, and $\text{Cu}_5\text{I}_7(\text{L}_4)_2$ crystals. Thus, no symmetry elements can be found for the $\text{Cu}_5\text{I}_7(\text{L}_2)_2$ and $\text{Cu}_5\text{I}_7(\text{L}_4)_2$ crystals of which space group $P1$ is the lowest degree of symmetry. For $\text{Cu}_4\text{I}_4(\text{L}_1)_4^{4+} \text{Cu}_8\text{I}_{12}^{4-}$ crystal, 4_2 screw axis along with c axis is one of the symmetry elements, which is shown by Replied Fig. 11. However, the helical

orientation cannot be judged for the 4_2 screw axis. But for the $\text{Cu}_5\text{I}_7(\text{L}_3)_2$ crystal, the symmetry operation based on 3_2 screw axis make the helical arrangement can be verified, which is M helix along the c axis (Replied Fig. 12). We have added the discussions and presentations about the spatial arrangements of clusters in the revised manuscript (Page 6, line 113) and Supporting Information (Supplementary Fig. 11-16).

Replied Fig. 11. 4-Fold screw axis of $\text{Cu}_4\text{I}_4(\text{L}_1)_4^{4+}$ $\text{Cu}_8\text{I}_{12}^{4-}$ hybrid clusters in crystal lattices. Color scheme: Cu, red; I, yellow; N, blue; C, Gray.

Replied Fig. 12. 3-Fold screw axis of $\text{Cu}_5\text{I}_7(\text{L}_4)_2$ hybrid clusters in crystal lattices. Color scheme: Cu, red; I, yellow; N, blue; C, Gray.

Comment 2): "To explain the transfer of chirality from ligand to the cluster and then to the phosphor powder, the CD spectra of ligands should be provided to compare

with single crystal powders and polycrystals.”

Reply: We thank the reviewer for this valuable suggestion. We have supplied the CD spectra of four chiral ligands. As shown in Replied Fig. 13, the CD signals of ligands are different from the corresponding single crystal powders and polycrystals, which imply the chirality transfer from ligands to the crystals and the chiral ligands induce the polycrystals to chiral morphology and chiroptical properties. *We have added the CD spectra of ligands in the revised Supporting Information (Supplementary Fig. 26).*

Replied Fig. 13. CD spectra of ligands **L₁**, **L₂**, **L₃**, and **L₄**.

Comment 3): “It is very interesting to see that different ligand and cluster design results in different performance in CPL such as photoluminescence and glum. If the authors can provide some summary and discussions on this structural design-optical performance correlations, that will further strengthen the mechanism illustration in this work.”

Reply: We gratefully thank the reviewer for the thoughtful comments. There are close associations between structural design and chiroptical performance. We present some guiding roles for the ligand and cluster design of high performance CPL. First, the alkyl chain length of the ligand plays a significant role in the luminescent efficiency and dissymmetry factors. To be specific, longer alkyl chains, such as L_3 and L_4 , will give rise to lower PLQYs. But short alkyl chains, like L_2 , will lead to the compact aggregation of nanoparticles, which cannot show an excellent helical morphology and glum. Second, the position of the chiral center in the alkyl chain is also important in the performance of final assembled hierarchically structured polycrystals. If the chiral center is away from the terminal of an alkyl chain, it will not lead to efficient chiral regulation in the crystallization process. Therefore, the chiroptical performance of hierarchical polycrystals is highly dependent on the structure of ligands. To obtain excellent CPL performance materials through the BNCC process, it is better to design a ligand that has an appropriate chain length and a chiral center on the terminal of the alkyl chain. *We have added this discussion and summary in the revised manuscript (Page 13, line 275).*

Comment 4): “The CCDC deposition numbers of all crystals should be listed in the manuscript for the sake of regulation.”

Reply: We thank the reviewer for this suggestion. The CCDC numbers have been added in the Data availability of the revised manuscript, which is 2125284, 2125285, 2125286, 2125291 for $Cu_4I_4(L_1)_4^{4+}$, $Cu_8I_{12}^{4+}$, $Cu_5I_7(L_2)_2$, $Cu_5I_7(L_3)_2$, and $Cu_5I_7(L_4)_2$, respectively.

Comment 5): “The word ‘well-defined’ at line 98, page 4 should be ‘well-designed’; the words ‘was still existed’ in line 201, page 8 should be ‘still exist’; the word ‘synthesized’ in line 338, page 15 should be ‘synthesize’.”

Reply: We thank the reviewer for the careful check. *We have corrected these improper writings in the revised manuscript (Page 4, line 67; page 9, line 174; page 15, line 317).*

REVIEWERS' COMMENTS

Reviewer #1 (Remarks to the Author):

The authors have well-addressed all the comments. The manuscript can be published without change.

Reviewer #2 (Remarks to the Author):

The authors have adequately revised their manuscript accordingly. The quality of the manuscript has been improved after the revision.

Reviewer #3 (Remarks to the Author):

The authors have revised the manuscript carefully according to the comments and suggestions of the reviewers, so publication is now recommended.